# An optogenetic toolbox for unbiased discovery of functionally connected cells in neural circuits

Dominique Förster[1], Marco Dal Maschio[1], Eva Laurell[1] & Herwig Baier[1]

Optical imaging approaches have revolutionized our ability to monitor neural network dynamics, but by themselves are unable to link a neuron's activity to its functional connectivity. We present a versatile genetic toolbox, termed 'Optobow', for all-optical discovery of excitatory connections in vivo. By combining the Gal4-UAS system with Cre/lox recombination, we target the optogenetic actuator ChrimsonR and the sensor GCaMP6 to stochastically labeled, nonoverlapping and sparse subsets of neurons. Photostimulation of single cells using two-photon computer-generated holography evokes calcium responses in downstream neurons. Morphological reconstruction of neurite arbors, response latencies and localization of presynaptic markers suggest that some neuron pairs recorded here are directly connected, while others are two or more synapses apart from each other. With this toolbox, we discover wiring principles between specific cell types in the larval zebrafish tectum. Optobow should be useful for identification and manipulation of networks of interconnected neurons, even in dense neural tissues.

[1] Department Genes—Circuits—Behavior, Max Planck Institute of Neurobiology, Am Klopferspitz 18, 82152 Martinsried, Germany. Correspondence and requests for materials should be addressed to H.B. (email: hbaier@neuro.mpg.de)

Neuronal circuits are composed of functionally distinct cells with specific morphologies, including characteristic axonal and dendritic arborization patterns. Neuronal structure in turn is related to synaptic specificity, which may differ greatly even in adjacent neurons[1–3]. While technology to record the activity of many neurons simultaneously has advanced rapidly[4–6], it remains challenging to decipher both functional and structural interconnectivity of cell types in a systematic fashion. Two approaches have been used to overcome this limitation: (1) electrophysiological recordings from randomly chosen pairs of neurons, followed by dye filling and inspection of neuronal morphologies[1], and (2) serial electron microscopic reconstructions of wires and synapses in a volume of brain tissue (connectomics)[2, 7]. Both of these approaches are laborious and not easily scalable. Moreover, paired recordings are intrinsically biased to certain cell types and not suitable to map long-range connectivity, while connectomic approaches are limited to postmortem, fixed tissue, thus lacking information about the functional dynamics of the network. Additionally, small organic and inorganic substances, as well as neurotropic viruses can be exploited for transsynaptic tracing of connected neurons in anterograde or retrograde directions[8]. Even though viral tracing technology allows the identification of monosynaptic connectivity, all pre- or postsynaptic neurons are labeled simultaneously, preventing their morphological characterization in dense circuits[9, 10].

Targeted activation of light-gated ion channels, such as channelrhodopsin[11], is slated to accelerate the discovery of functional connections provided that its effect on connected cells can be recorded simultaneously. Pioneering work has combined optogenetics with electrophysiological recordings to map long-range neural circuits in brain slices[12]. An all-optical approach might complement electrophysiology with imaging of neuronal activity in functionally connected cells, as monitored by a genetically encoded calcium sensor, such as GCaMP[13–15]. Recent studies achieved simultaneous manipulation and recording with cellular resolution by using spectrally separated actuators and sensors[16–19]. However, due to co-expression of optogenetic actuator and indicator in all neurons of a given population, this work has not allowed a detailed morphological classification of the functionally connected neurons and has been restricted to spatially distant cells.

Here we propose an all-optical toolbox, comprising cellular-resolution optogenetics and a new set of genetically engineered zebrafish. Our Optobow approach circumvents many of the problems associated with other methods and enables systematic discovery of excitatory circuit components in the intact brain. Modifications of the Optobow toolbox may be used not only for circuit mapping, but also for studies of synaptic plasticity or the analysis of cell-restricted gene function in the living brain.

## Results

**The Optobow concept and its components.** We sought to develop a genetic toolkit for all-optical mapping of neuronal connectivity to which the following attributes apply. First, optogenetic actuator and calcium sensor should not be co-expressed in the same cell. This minimizes the direct photostimulation of neuronal processes of non-targeted cells or the alteration of their membrane potential due to the imaging process itself (see example in Fig. 1e). Second, expression of actuator and sensor should be stochastic to allow unbiased probing of connectivity. Third, expression patterns should be sparse to facilitate unambiguous identification of cellular morphologies and, again, to prevent unintended photostimulation of non-targeted cells. To develop such a transgenic labeling kit, we made use of the

'Brainbow' configuration[20]. This enabled us to separate the actuator and sensor by mutually exclusive pairs of lox sites inside a single transgene (Fig. 1a and Supplementary Fig. 1). We further placed this construct, which we termed Optobow, under control of the Gal4/UAS system to introduce genetic specificity.

For the first version of Optobow (Optobow-c, cytoplasmic), we selected an actuator-sensor pair that shows spectrally well-separated excitation bands. Besides successful application of C1V1$_T$ and ReaChR channels, we found ChrimsonR to be our actuator of choice, as it minimizes the overlap with the action spectra of GCaMP6 calcium indicators[13, 21–23]. Initially, we confirmed that ChrimsonR is functional in zebrafish using an established behavior protocol. Optic fiber stimulation of premotor neurons expressing ChrimsonR with 638 nm light provided sufficient activation to elicit a robust steering movement of the zebrafish tail[24] (Supplementary Fig. 2 and Supplementary Movie 1).

We generated several transgenic zebrafish lines with different levels of genetic variegation to adjust the sparseness of Optobow expression for each experiment at hand (Supplementary Fig. 3). Gal4-dependent expression of a single copy of Optobow-c in the optic tectum of larval zebrafish and neural-specific expression of Cre resulted in sparse, nonoverlapping labeling of random cells with either ChrimsonR or GCaMP6f, respectively (Fig. 1b, c and Supplementary Fig. 1). In the absence of Cre-mediated recombination, cells were marked by mCerulean expression, which served as a background label of the cell population of interest. With this construct, on average about double as many cells end up expressing GCaMP6f compared to ChrimsonR (ratio of GCaMP6f/ChrimsonR cells = 2.23; $n = 10$ fish). This feature, which most probably originates from an intrinsic preference of Cre for the canonical loxP sites, is useful, as it sparsens the cells that can be stimulated relative to the cells from which recordings can be made.

**Mapping of connected neurons with Optobow-c.** To probe functional connectivity at high resolution, we used a setup that incorporates two independent femtosecond-pulsed lasers: one, tuned to 920 nm for two-photon imaging of GCaMP fluorescence, and a second source for ChrimsonR photostimulation (Fig. 1d), coupled to a custom phase-modulation optical train to achieve spatially selective illumination profiles based on two-photon computer-generated holography[25] (Supplementary Fig. 4). Two excitation wavelengths were found to be effective for ChrimsonR stimulation, 760 and 1020 nm, resulting in reliable triggering of calcium transients locked to the stimulation protocol in the targeted cells (Supplementary Fig. 4). Switching to 920 nm, while keeping the same power density, did not result in significant calcium changes with respect to baseline levels. Even though high-resolution photostimulation methods allow to reduce the excitation profile for stimulation to dimensions close to cell size (Supplementary Fig. 4), dense expression of the actuator could still lead to inadvertent photostimulation of ChrimsonR-expressing neurites close to the targeted cell[15, 25] (Supplementary Fig. 4). Thus, sparse expression of the actuator is beneficial to minimize this risk and, at the same time, allows morphological reconstructions.

In the larval zebrafish brain, the tight packing of neurons poses a particular challenge for attempts to map functional connectivity. In addition, neuronal cell bodies are relatively small, averaging about 5 μm in diameter. For instance, in single image planes of the tectum, the largest midbrain area, cell bodies of periventricular neurons (PVNs) show a local density of 4.3 cells per 100 μm$^2$ ($n = 69$). These neurons send their dendrites into a densely packed neuropil region, where they arborize in different layers and receive synaptic inputs from either retinal ganglion cell

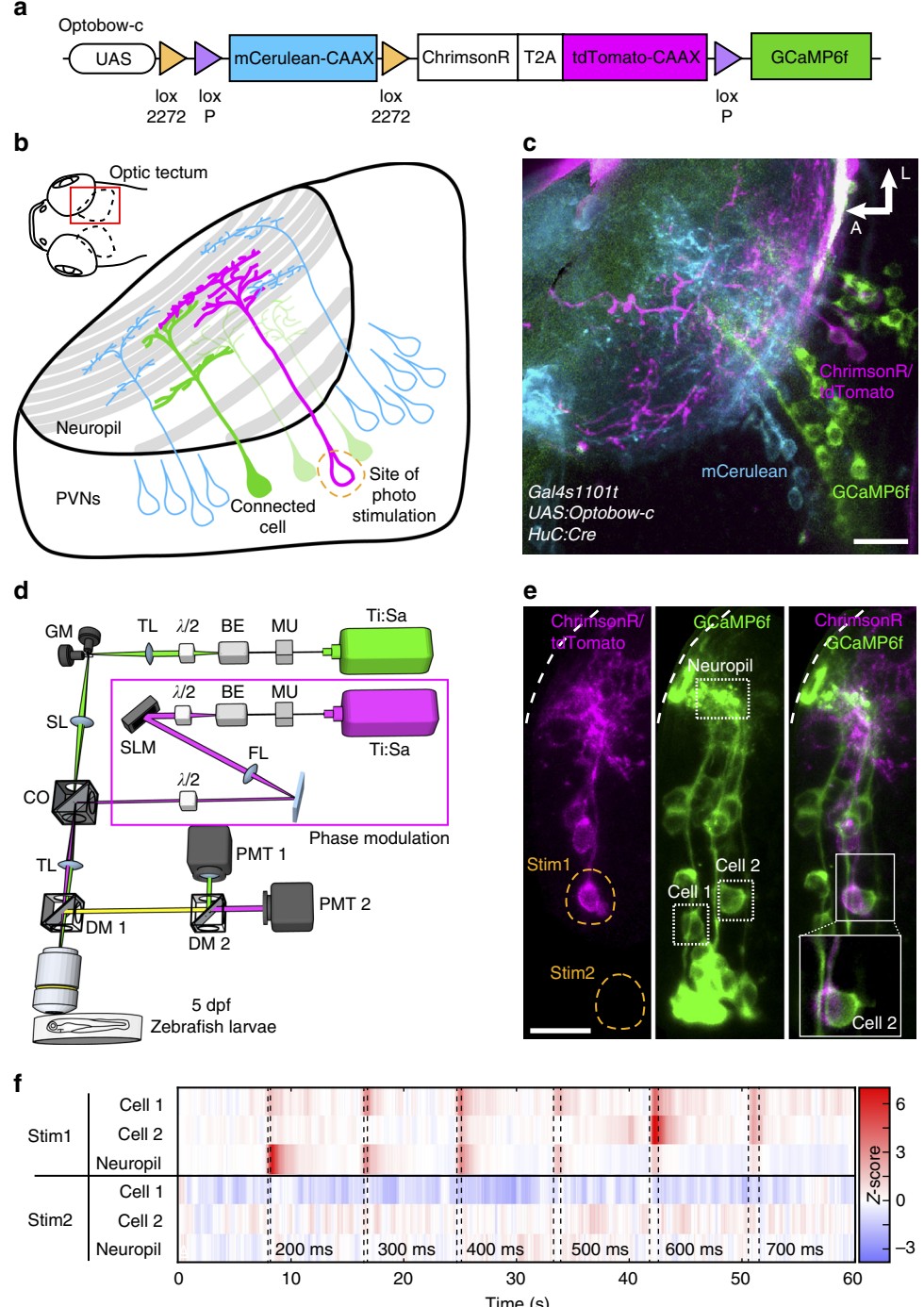

**Fig. 1** Optobow allows all-optical mapping of connected neurons. **a** Design of the Optobow-c construct. mCerulean and tdTomato are both membrane-targeted (CAAX motif). **b** Sketch for Optobow-c expression in the zebrafish optic tectum. Different cell types are indicated by their arborization patterns in the tectal neuropil. Addition of Cre by transient injections results in random expression of either ChrimsonR (labeled by tdTomato, *magenta*) or GCaMP6f (*green*). Unrecombined cells are labeled by mCerulean (*blue*). Upon photostimulation of a single ChrimsonR cell (*dashed orange outline*) connected cells will be highlighted by a rise in GCaMP fluorescence (*dark green cell*). **c** In vivo tectal expression of Optobow-c. Scale bar, 20 μm. **d** Microscope setup for all-optical connectivity mapping. Two independent infrared-pulsed lasers are used to photostimulate ChrimsonR cells (*magenta*) and to image GCaMP fluorescence (*green*), respectively. A spatial light modulator (SLM) allows precisely targeted stimulation of single cells. See methods for abbreviations. **e** Dorsal view of tectal Optobow-c expression. Single fluorescent channels show a cluster of three ChrimsonR- (*left*) and 18 GCaMP6f-expressing cells (*middle*). Photostimulation was either confined to a single ChrimsonR cell (Stim1, *dashed orange outline*) or to a control region of equal size (Stim2). *White dashed line* marks skin. *Dotted rectangles* show regions of calcium imaging during photostimulation. A close-up showing a single confocal plane (z = 2.5 μm) of the photostimulated region (Stim1) is shown in the merge on the *right*. Despite the close proximity of the ChrimsonR and the GCaMP cell#2, exclusive expression of either component allows restricted stimulation of the ChrimsonR-expressing cell. *Scale bar*, 15 μm. **f** Calcium transients acquired at 10 fps shown as *Z*-scores, obtained simultaneously from the regions indicated in **e**. Photostimulation events are highlighted by *dashed lines* with stimulation lengths indicated below. Off-target stimulation did not result in significant GCaMP activity (Stim2)

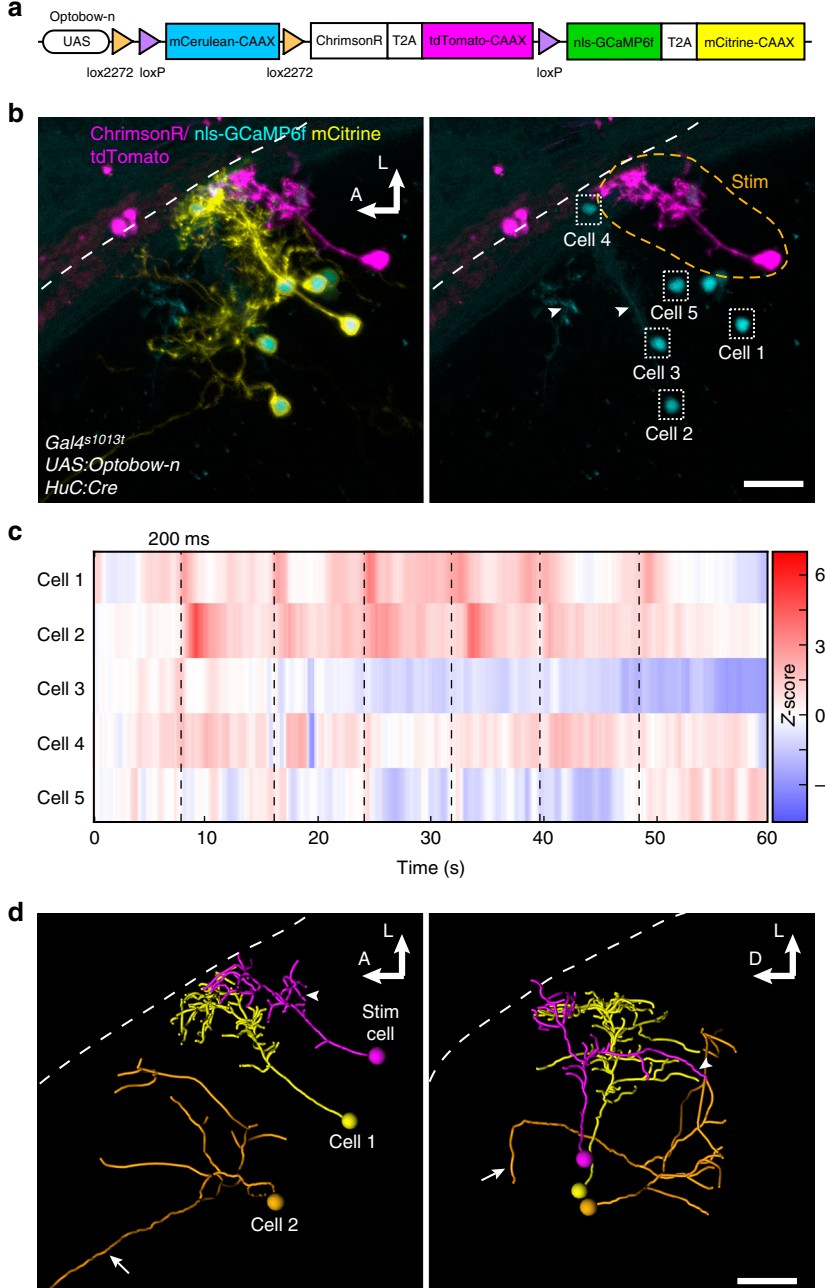

**Fig. 2** Simultaneous connectivity mapping and morphological analysis with Optobow-n. **a** Schematic of the Optobow-n construct. **b** Tectal-specific expression of Optobow-n. A single ChrimsonR-expressing cell (*magenta*) is surrounded by seven nls-GCaMP6f-expressing cells, membrane-labeled by mCitrine (*yellow*). Localization of GCaMP6f (*cyan*) appears completely restricted to the nucleus. *White dashed line* indicates skin. *Orange dashed line* in two-channel merge (*right* image) marks photostimulated region. *Dotted rectangles* show regions of calcium imaging during photostimulation. *Arrowheads* show mCerulean signals in non-recombined cells. Scale bar, 20 μm. **c** Calcium transients (*Z*-scores) acquired at 10 fps from the regions annotated in **b**. Photostimulation events of 200 ms are marked by *dashed lines*. While calcium activity of cell#1 is tightly coupled to the photostimulation, activity of cell#2 appears slightly delayed. No significant calcium responses were detectable in other neighbouring cells (#3–5). **d** Three-dimensional tracings of the photostimulated cell (*magenta*) and the two responding cells. Dorsal view is shown on the *left* and transverse view on the *right*. The stimulated cell is a bistratified projection neuron with a descending axon (*arrowhead*). Cell#1 is a bistratified periventricular interneuron and cell#2 is a non-stratified projection neuron, which sends an axon to the intertectal commissure (*arrows*). Scale bar, 20 μm

(RGC) axons, other afferents, or the axons of tectal interneurons[26]. The axons of PVNs extend from the same stem neurite as the dendrites and either synapse on other tectal neurons or leave the tectum to form connections with areas in the diencephalon, midbrain or hindbrain.

With Optobow-c, we achieved non-overlapping and sparse expression of actuator and sensor in the PVNs of the tectum,

sometimes in directly adjacent cells (Fig. 1e). When we targeted the photostimulation profile to single ChrimsonR-expressing cells, we often observed fluorescence changes in GCaMP6f-positive cells (Fig. 1e, f; Supplementary Fig. 5). In the experiment shown in Fig. 1f, two cells showed activity locked to the photostimulation event; other neurons were not responsive or showed inconsistent activity. Signals recorded in the neuropil

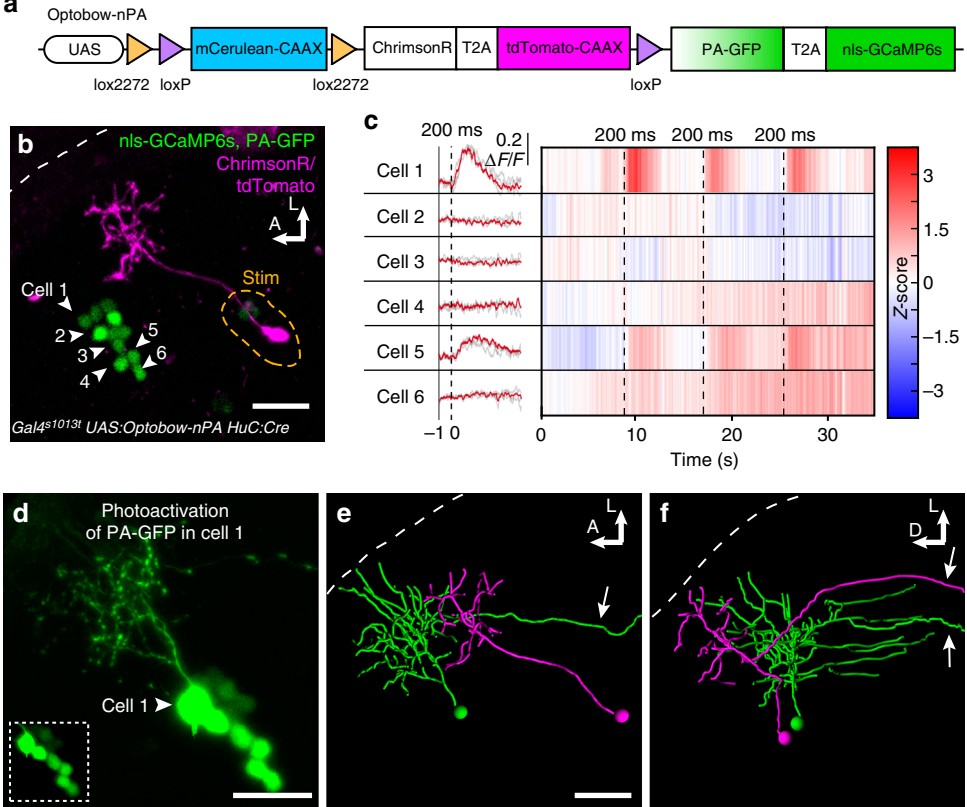

**Fig. 3** PA-GFP reveals morphologies of functionally connected cells. **a** In Optobow-nPA, all 'nls-GCaMP6s' cells co-express PA-GFP. **b** Optobow-nPA expression in the optic tectum. A single ChrimsonR-expressing cell (*magenta*) was photostimulated (*orange dashed line*) and GCaMP fluorescence was monitored in six neighbouring cells by line scans across the nuclei. Note that PA-GFP is not detectable before its activation. Scale bar, 20 μm. **c** Calcium measurements acquired at 250 fps for the GCaMP cells annotated in **b**. Raw (*grey*) and averaged Δ*F/F* responses (*red*) are shown on the *left*, heat maps for *Z*-scores are on the *right*. Photostimulation epochs of 200 ms are indicated by dashed lines. Cell#1 and #5 showed reliable calcium responses upon photostimulation. **d** Close-up of cell#1 after photoactivation of PA-GFP. A less saturated, single slice of the cell body region shown in the *lower left* demonstrates exclusive photoactivation of cell#1. Scale bar, 20 μm. **e**, **f** Three-dimensional filament reconstruction of the presynaptic cell (*magenta*) and cell#1 (*green*) in dorsal view (**e**) and transverse view (**f**). The presynaptic cell is a bistratified projection neuron, and cell#1 is a non-stratified projection neuron. Descending projection axons are marked (*arrows*). Scale bar, 20 μm

region were highly correlated with the photostimulation events, although dense neurite clustering and the diffuse cytoplasmic GCaMP6f fluorescence prevented a clear identification of the corresponding cell bodies. No calcium transients were detectable during off-target stimulations. We conclude that Optobow-c is suited for all-optical identification of direct or indirect, functional connections in densely packed brain areas.

**Analysis of connectivity and morphology with Optobow-n.** Cytoplasmic expression of the sensor, as in Optobow-c, is the method of choice when calcium responses need to be monitored in the processes of postsynaptic cells, such as in neuropil areas. However, it is less suitable for morphological reconstruction of single cells. We therefore re-engineered the reporter segment of the construct, introducing a strong labeling of the cellular membranes by the bright fluorescent protein mCitrine and a nuclear localization signal (nls) to restrict GCaMP6f fluorescence exclusively to the nucleus (Fig. 2a). This transgene, termed Optobow-n (nuclear), allowed morphological discrimination of individual GCaMP6f-expressing cells, facilitated segmentation of cellular signals, and improved the signal-to-noise ratio of calcium transients, especially of the rare neurons that have their cell bodies inside the neuropil (superficial interneurons, SINs; Fig. 2b, cell#4).

We first compared the dynamics of nuclear and cytoplasmic GCaMP6 versions upon photostimulation of co-expressed ChrimsonR (Supplementary Fig. 6). While the basal levels, relative fluorescence change, and decay times were not significantly different, we found that the fluorescence rise time was slightly longer for nls-GCaMP6 versions (Supplementary Fig. 6), similar to previous reports[27, 28]. When we used Optobow-n to map connectivity in the optic tectum, nls-GCaMP6f reliably reported calcium activity in directly or indirectly connected cells (Fig. 2c and Supplementary Fig. 5). Further, the strong mCitrine membrane label allowed us to trace the fine axons and dendrites of connected and unresponsive cells, even of periventricular projection neurons, whose long axons project from the optic tectum to other brain areas (Fig. 2d and Supplementary Movie 2). Close anatomical proximity alone was a poor predictor of functional connectivity to the stimulated cell (Fig. 2b, cell#3–5). These results demonstrate that Optobow-n is a valuable tool for all-optical identification of neuronal partners of one functional circuit, particularly for experiments in which sparse labeling of neurons is possible and instant information about neuronal morphology is desirable.

**Reconstruction of neurons in dense circuits with Optobow-nPA.** To increase the yield of Optobow experiments,

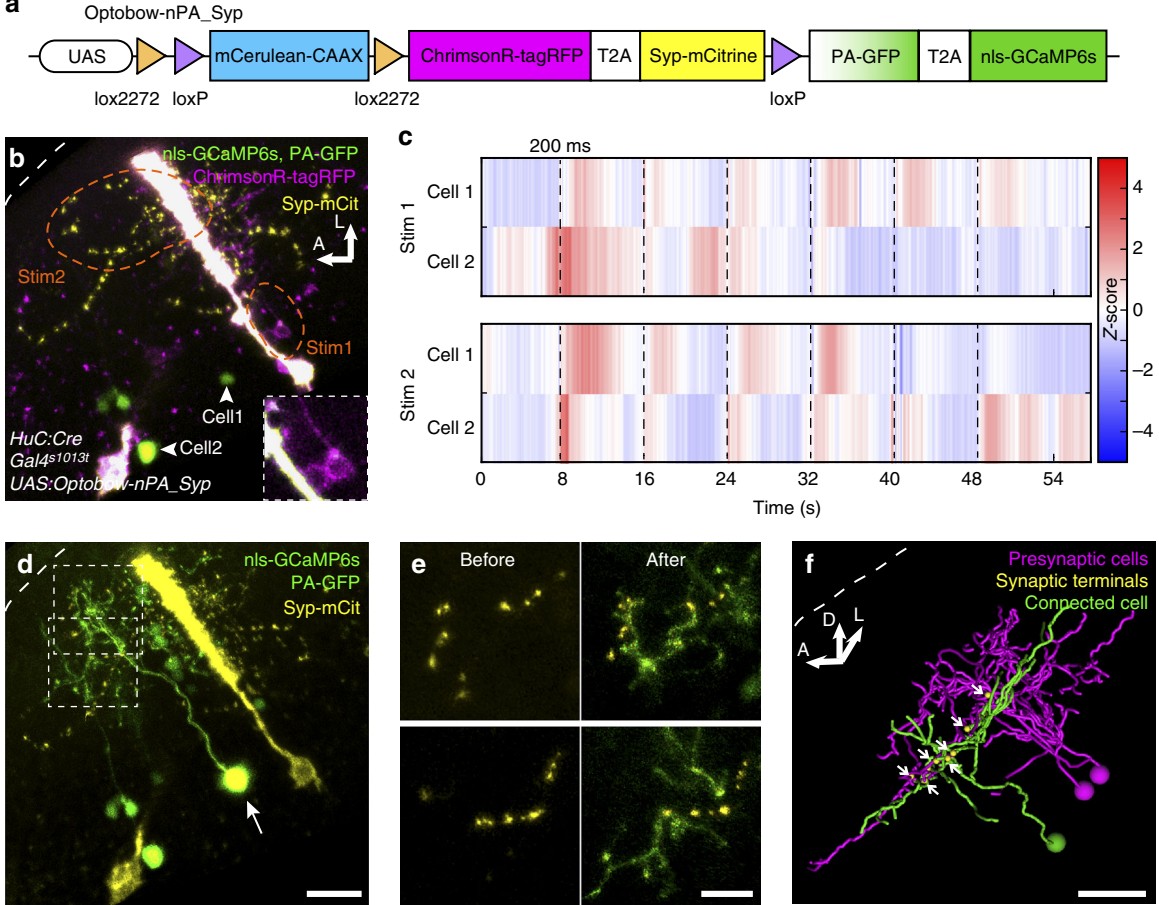

**Fig. 4** Optobow-nPA_Syp reveals potential synaptic contacts. **a** In Optobow-nPA_Syp, ChrimsonR-expressing cells are co-labeled by Synaptophysin-mCitrine. **b** Tectal-specific expression of Optobow-nPA_Syp. Cell bodies of two ChrimsonR-expressing cells (*magenta*, close-up in *lower right*, Stim1) or neuropil regions (Stim2), respectively, were stimulated (*orange dashed line*), and activity of two nls-GCaMP6s-expressing cells (*arrowheads*) was monitored. Note that a radial glia cell overexpressing ChrimsonR-tagRFP and Syp-mCitrine appears in *white*. **c** Z-scores of calcium measurements for cell#1 and cell#2 during cell body (Stim1) or neuropil stimulations (Stim2). Photostimulation epochs of 200 ms are indicated by *dashed lines*. Cell#1 shows high response reliability. While cell#2 shows spontaneous activity during Stim1, its response correlates to neuropil stimulations suggesting that additional ChrimsonR-labeled cells were activated by Stim2. **d** Photoactivation of PA-GFP in cell#1 (*arrow*). Spectral unmixing was used to separate PA-GFP/GCaMP (*green*) from mCitrine signals (*yellow*; see Methods section). Scale bar, 20 μm. **e** Close-up on single confocal slices of the regions marked in **d** before and after PA-GFP photoactivation. Scale bar, 10 μm. **f** Three-dimensional filament reconstruction of presynaptic (*magenta*) and connected cells (*green*) in transverse view. Potential synapses, shown in *yellow* (*arrows*), are restricted to a single tectal layer. *Scale bar*, 20 μm

simultaneous monitoring of several postsynaptic cells would be favourable. To enable unambiguous discrimination of single cell morphologies in a densely packed tissue, we engineered a variant of the construct, named Optobow-nPA, with photoactivatable GFP (PA-GFP) replacing the permanent membrane marker mCitrine (Fig. 3a). Combining the advantages of a selectively inducible marker by targeted illumination, together with a nuclear segregated calcium signal, allowed us to highlight specifically only those single cells, which show activity upon photostimulation of ChrimsonR-expressing cells. In this new design, we also introduced nls-GCaMP6s because its higher signal-to-noise ratio allowed us to increase the acquisition rate up to 350 Hz by scanning in a random access mode, while keeping the imaging power far below that necessary for inadvertent activation of PA-GFP or for photostimulation of ChrimsonR-expressing axons (Fig. 3b). In this regime, co-expression of cytoplasmic PA-GFP did not interfere with the dynamics of nuclear GCaMP6s signals detected upon ChrimsonR photostimulation (rise time nls-GCaMP6s = 1133 ± 58 ms; nls-GCaMP6s + PA-GFP = 1126 ± 84 ms; $n = 16$; $P = 0.8$; Fig. 3c and Supplementary Fig. 5). Further,

our spatially selective stimulation scheme prevented unspecific PA-GFP photoactivation during ChrimsonR stimulation performed at 760 nm. Photoactivation of PA-GFP could be performed precisely and exclusively in the cell of interest, and within few minutes its morphology was revealed (Fig. 3d–f).

**Anatomical and quantitative characterization of connections.** Some of our functionally identified connections may be carried by direct chemical synapses. If so, presynaptic markers, such as synaptophysin (see Meyer and Smith[29]), should be localized to regions in which the axons of the photostimulated cell overlap with the dendrites of the activated cell. Using an extended version of Optobow-nPA, we achieved co-labeling of ChrimsonR-expressing cells with Synaptophysin-mCitrine (Syp-mCit) (Fig. 4a, b). Spectral unmixing allowed us to identify Syp-mCit punctae that were in close proximity (<1 μm) to postsynaptic PA-GFP-labeled dendrites. Thus, based on cellular proximity and synaptic marker staining, the use of Optobow-nPA_Syp may suggest the existence of direct connectivity between the functionally identified partners (Fig. 4b–f and Supplementary Fig. 5).

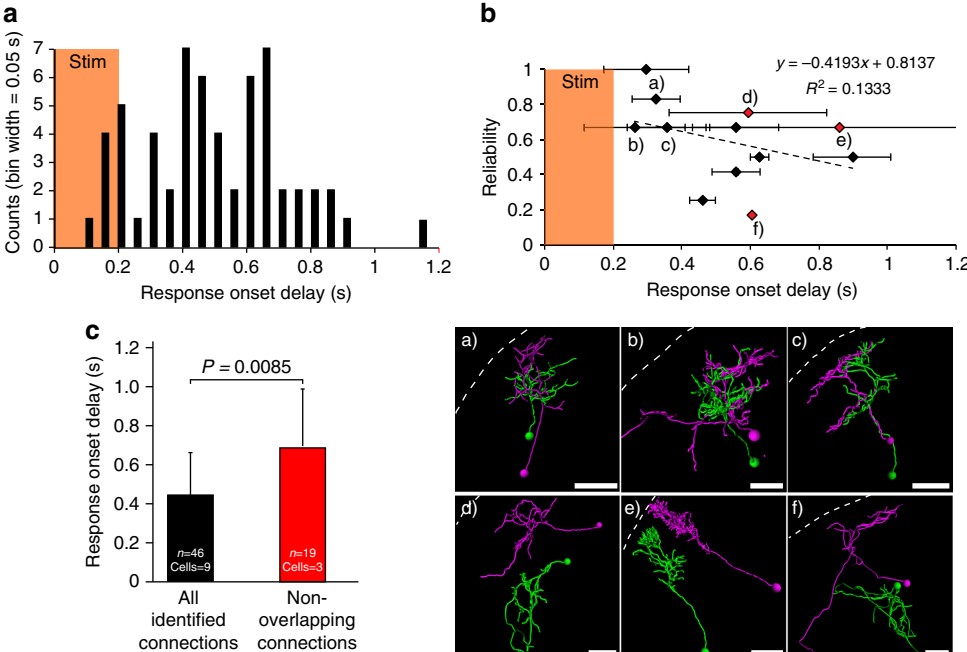

**Fig. 5** Quantification of response latencies and reliabilities. All data are derived from 5 dpf larvae expressing Optobow-nPA, using a sampling rate for calcium signals of <4.5 ms per frame. **a** Distribution of all response latencies. Events are grouped in temporal bins of 50 ms. *Orange* shaded region marks 200 ms of ChrimsonR photostimulation. Number of trials $n = 65$, number of cells = 12. **b** Individual cell response reliability vs. response onset latency. For every cell, the average response onset is represented along the $X$ axis, while the ratio responses per trials is plotted in the $Y$ axis. Dashed line represents a linear regression model of the data to evaluate the degree of correlation (coefficient of determination is 0.13329). Morphological analysis shown below indicates overlapping (a–c) or non-overlapping (d–f) neurite arbors of the functionally identified cell pairs. Error bars are SD. *Scale bar*, 20 μm. **c** Average response latencies of three non-overlapping cells (*red data points* in **b**) compared to all other identified responding cells. Error bars are SD

Many of our morphologically reconstructed cell pairs showed overlapping neurite arbors, providing an opportunity for the formation of direct contacts between presynaptic axons and postsynaptic dendrites. However, we also identified pairs of connected cells in our sample, whose neurite arbors did not overlap, suggesting polysynaptic transmission. We reasoned that physical separation might correlate with the latency of evoked responses. By sampling GCaMP6 signals at a frequency of several hundred Hz, we found that cell pairs that did not overlap showed generally longer latencies compared to all other identified cell pairs (Fig. 5). Based on this analysis, we also noticed a slight inverse correlation between the latency and the reliability of responses. Thus, our method is able to reveal chains of excitation over several synapses.

**Detection of long-distance connectivity.** So far, we showed that Optobow-nPA facilitates mapping and morphological classification of directly or indirectly connected neurons in densely packed areas such as the tectum, where excitatory connectivity is expected to be sparse. We asked if the Optobow approach can also be applied to probe long-range connections between neurons. The *Gal4s1013t* driver, best known for labeling tectal neurons[30], frequently also labels a subset of RGCs, which project their axons to the tectum. Sparse and random Optobow expression in both RGCs and PVNs allowed targeted photostimulation of a single RGC axon expressing ChrimsonR, while monitoring several PVNs labeled by nls-GCaMP6s (Fig. 6a–c and Supplementary Fig. 5). Ultimately, the connected neuronal cell types were classified by PA-GFP photoactivation and anti-tdTomato immunostainings (to reveal the dendritic pattern in the retina). The example in Fig. 6 shows the functional interaction of an OFF-type RGC, whose axon is entering the superficial tectal layer, and a bistratified periventricular interneuron (Fig. 6d, e). This finding

suggests that Optobow-nPA might be used to identify and characterize different functional types of neurons, which are connected over long distances.

**Discovery of novel connections within the tectum.** The reconstruction of connectivity patterns using Optobow-nPA allowed us to demonstrate previously unknown, functional interactions between neuronal cell types in the optic tectum. Every formerly described tectal cell morphology appeared at least once in our collection, suggesting that the Optobow approach is not biased to particular cell types (Figs 2, 3, 4, 5, 6, 7 and 8). Optobow analysis even led to the identification of novel tectal cell types and their interactions, including tristratified periventricular interneurons and tristratified projection neurons (Figs 7 and 8). Further, we discovered that tectal projection neurons, which send long efferents to premotor areas or to the contralateral tectum, are directly or indirectly connected to neighbouring interneurons, or to other projection neurons (Figs 2, 3, 7 and 8). We are not aware of another method that would have revealed this surprising connectivity principle in the fish tectum. While their function is unknown, we speculate that these connections may provide a substrate for efference copies that refine, enhance, or coordinate tectal output to other brain areas.

**Anatomical registration of interconnected cell pairs.** We placed neuronal pairs, belonging to individual functional circuits, into one reference brain[31, 32], using a state-of-the-art image registration technique, i.e., the Computational Morphometry Toolkit (CMTK) (Fig. 8 and Supplementary Movie 3). The fully automated algorithm was applied to four-channel confocal stacks of immunostained fish, including a ubiquitous neuronal marker (Elavl3/HuC) as a reference. The registration allowed a direct comparison of the position of the individual cell pairs in the

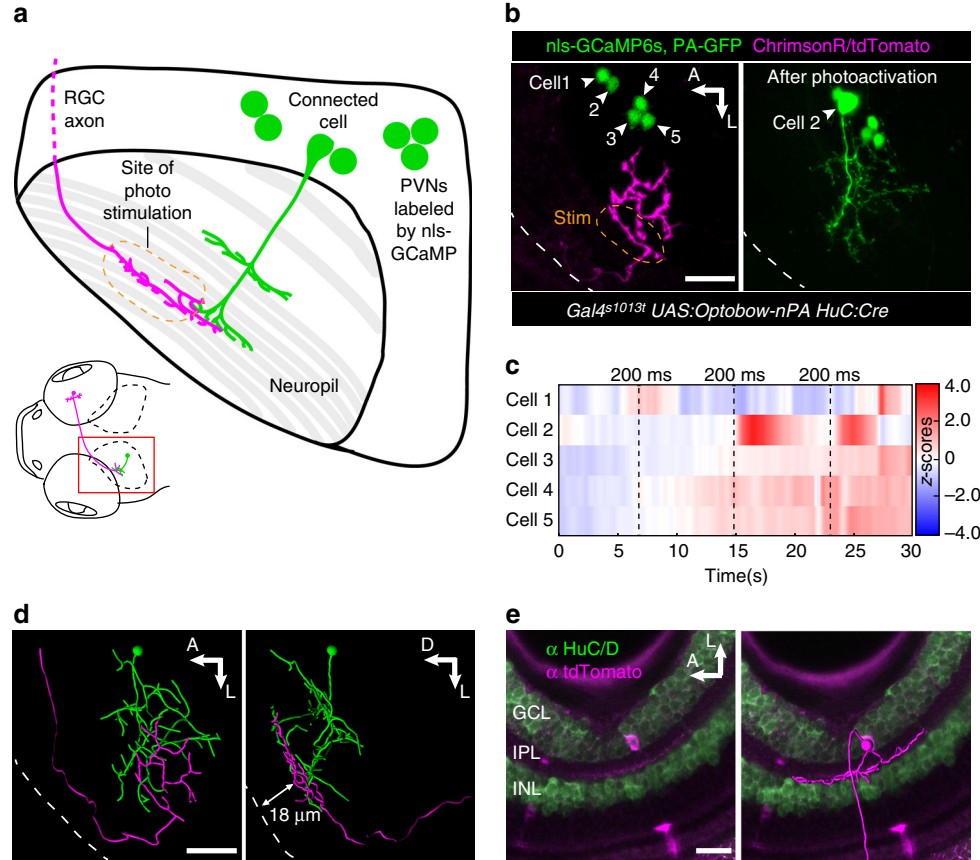

**Fig. 6** Optobow analysis can be expanded to study long-range connectivity of different cell types. **a** Sketch simplifying the connection of an RGC axon with a PVN in the optic tectum. **b** Live expression of Optobow-nPA. A single RGC axon (*magenta*) and five PVN nuclei are labeled (*green, left*). White dashed lines mark the skin. Upon identification of the connected cell#2, PA-GFP was photoactivated (*right*). Scale bar, 20 μm. **c** Calcium traces (*Z*-scores) for the cells annotated in **b**. Photostimulation epochs of 200 ms are indicated by *dashed lines*. Only cell#2 showed responses upon photostimulation. **d** Filament reconstructions in dorsal view (*left*) and transverse view (*right*). The RGC axon terminates in the stratum fibrosum et griseum superficiale (SFGS) layer 1 (18–20 μm distance from skin), where it contacts the dendrites of the bistratified PVIN (cell#2). Scale bar, 20 μm. **e** Immunostainings against HuC/D (*green*) and tdTomato (*magenta*) reveal the dendritic pattern of the activated RGC in the retina. A filament reconstruction is shown on the *right*. Its monostratified dendrites exclusively target the OFF layer of the IPL. Scale bar, 20 μm. GCL, ganglion cell layer; INL, inner nuclear layer; IPL, inner plexiform layer; PVIN, periventricular interneuron

midbrain and a more detailed visualization and comparison of tectofugal projections. In the future, additional alignments of reference patterns, like the innervation strata in the tectal neuropil, could potentially allow even more detailed characterization of dendritic and axonal arbors of tectal cell types (Supplementary Fig. 7). In conclusion, this final registration step successfully links the morphologies of single neurons to the mesoscale architecture of the tectum and the tectorecipient areas.

## Discussion

Here we have devised an experimental workflow, including new genetic constructs and transgenic zebrafish lines, to reveal excitatory connectivity between pairs of neurons in dense neural circuits. We can see three general applications of the Optobow toolbox. First, as is the primary focus of our study here, Optobow can be used as an unbiased circuit mapping technique to demonstrate direct or indirect, functional connections between morphologically identifiable cells, especially in areas of the brain where electrophysiological recordings are difficult due to the small sizes and dense packing of neurons. Second, modifications of Optobow could be employed to follow the formation of synapses or the plastic changes in synaptic strength over periods of time that are prohibitive to in vivo electrophysiology. Such

modifications may bring into reach experiments that require long-term monitoring of one and the same connection during development or learning of a behaving animal[33]. Third, Optobow might be used in conjunction with targeted genetic or pharmacological perturbations whenever single cells need to be manipulated within a complex tissue. In fact, Optobow is well compatible with mosaic analyses in which interactions between neurons with distinct genotypes need to be determined in an intact brain.

The Optobow approach could be scaled up for a systematic, brain-wide survey of excitatory cell-type interconnectivity, possibly in conjunction with dense reconstruction techniques, like serial EM[2, 7] or expansion microscopy[34]. Optobow may provide functional validation of hypothetical synapses between morphological cell types and may help to correctly judge and then remove ambiguities resulting from imperfect staining of fixed tissue. Further, Optobow may assist the reconstruction of long-range axonal projections spanning different brain regions, which remains a challenging task for serial EM reconstructions. Like the ground-breaking Brainbow technique[20], the Optobow toolbox can be adapted to other genetic model organisms, like fruit fly or mouse.

Currently available optical approaches are poorly suited to demonstrate direct synaptic contacts owing to the sluggish

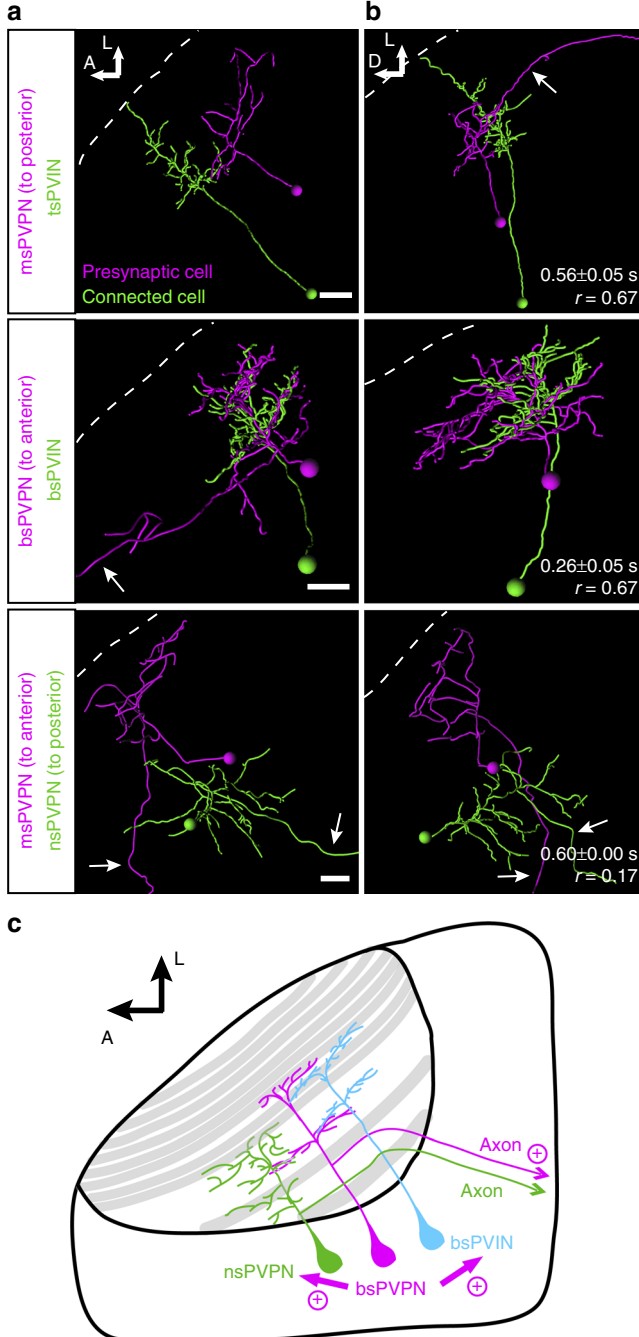

**Fig. 7** Optobow reveals novel excitatory connections of projection neurons within the optic tectum. **a, b** Morphological reconstructions of connected tectal cells, identified using Optobow-nPA, in dorsal view (**a**) and transverse view (**b**). Cell types are indicated on the *left*. *White dashed lines* mark the skin. *Arrows* point to projection axons leaving the neuropil. Values for response onset time (s) and response reliability (*r*) are shown in **b**. *Scale bar*, 15 μm. **c** Model for tectal connectivity of PVPNs. In addition to an axon leaving the tectum, PVPNs make functional, excitatory connections both with PVINs and other PVPNs. bs, bistratified; ms, monostratified; ns, non-stratified; PVIN, periventricular interneuron; PVPN, periventricular projection neuron; ts, tristratified

kinetics of both the actuator and the sensor. Moreover, GCaMP6 indicates the intracellular (cytoplasmic or nuclear) buildup of $Ca^{2+}$, which is a secondary or tertiary consequence of an action potential and therefore intrinsically delayed. Nevertheless, the shortest latencies we observed following photostimulation were

responses within the fastest, measurable range of the GCaMP6 indicator. In these cases, neurites of potentially pre- and post-synaptic cells tended to be in close anatomical proximity. Indeed, in some experiments, we confirmed the presence of chemical synapses between abutting neurites by the presence of the pre-synaptic marker Synaptophysin-mCitrine. A more direct and faster readout of neuronal activity would be membrane depolar-ization. However, the dynamic range of currently available genetically encoded voltage sensors may be too small for in vivo applications. Definitive evidence for monosynaptic connectivity might have to come in the future from Optobow mapping fol-lowed by correlated electron microscopy.

Currently, Optobow is able to reveal excitatory connections that are strong enough to cause action potentials in downstream cells. In rare cases, identified calcium responses may even be the result of strong postinhibitory rebound spiking. We envision that depolarizing or hyperpolarizing subthreshold events could, in the future, be visualized by voltage or neurotransmitter sensors[35, 36]. The modular design of Optobow enables further expansion of the toolkit, making this system flexible when improved actuators or sensors become available (Supplementary Fig. 1). Furthermore, to increase the chance of finding connected neurons, while main-taining sparse expression of the optogenetic actuator, Cre con-centration and *UAS:Optobow* constructs could be adjusted to the task at hand. With such improvements, novel experimental paradigms that determine, simultaneously, the contribution of a single neuron to network activity and behavior[37] may become possible.

## Methods

**Transgenic constructs and zebrafish lines**. The initial Optobow cassette was synthesized by Genscript (Piscataway, NJ, USA) and cloned into a *pTol2-14xUAS* vector. The open reading frames for ChrimsonR[23], ReaChR[22], C1V1$_T$[21], GCaMP6f, GCaMP6s[13], PA-GFP (gift from K. Svoboda), mCitrine (gift from M. Lin), zeb-rafish Synaptophysin (gift from M. Meyer), and the N-terminal nls[38] were PCR amplified and cloned into *pTol2-UAS:Optobow* to obtain different variants (Sup-plementary Fig. 1). Similarly, *GCaMP6f, GCaMP6s, nls, ChrimsonR,* and *tagRFP-T*[39] were PCR amplified to generate *pTol2-UAS:GCaMP6f, pTol2-UAS:nls-GCaMP6f, pTol2-UAS:nls-GCaMP6s,* and *pTol2-UAS:ChrimsonR-tagRFP-T,* respectively. The Cre coding sequence and the *HuC (elavl3)* promoter were PCR amplified from *pCR8GW-Cre-FRT-kan-FRT*[40] and from *pT2d-HuC:Gal4-VP16*[41], respectively, to clone *pTol2-HuC:Cre.* For the generation of *pTol2-elavl3:nls-GCaMP6s,* a longer variant of the *HuC* promoter was PCR amplified from a *HuC: GCaMP5G* plasmid (gift from M. Orger) and cloned into *pT2KXIGin*[42].

All animal procedures conformed to the institutional guidelines of the Max Planck Society and the local government (Regierung von Oberbayern). To generate transgenic lines, wild-type fish with a *Tüpfel long fin nacre (TLN)* background were injected at the one-cell stage with 25 ng μl$^{-1}$ plasmid DNA and 25 ng μl$^{-1}$ Tol2 transposase mRNA. The following transgenic lines were generated: *Tg(UAS:nls-GCaMP6f)mpn133*; *Tg(UAS:ChrimsonR-tagRFP)mpn115*; *Tg(elavl3:Cre)mpn403*; *Tg(elavl3:nls-GCaMP6s)mpn400*. For a list of transgenic Optobow lines, see Supplementary Fig. 3. Absence of co-expression of fluorescent markers suggests that all Optobow lines are single-copy insertions. All of the newly generated transgenic lines are viable and fertile, and pan-neuronal expression of the single optogenetic probes does not compromise cell health.

The following existing transgenic lines were used: *Et(−1.5hsp70l:Gal4-VP16) s1013t (=Gal4s1013t); Et(E1b:Gal4-VP16)s1101t (=Gal4s1101t); Et(−0.6hsp70l: Gal4-VP16)s1171t (=Gal4s1171t); Tg(isl2b:Gal4-VP16)zc65; Tg(UAS:GCaMP6s) mpn101.*

**Optobow zebrafish preparation**. *Gal4s1013t UAS:Optobow* transgenic fish were crossed to TLN wild-type fish, and embryos were raised in fish water. *HuC:Cre* plasmid DNA was injected into one-cell stage embryos at varying concentrations (2–25 ng μl$^{-1}$). For Optobow-nPA_Syp experiments, the *UAS:Optobow* construct was co-injected together with *HuC:Cre* at 25 ng μl$^{-1}$ concentrations. At 24 hpf, larvae were sorted at the fluorescence scope for sparse expression of green and red fluorescent proteins and carriers were transferred to E3 medium containing 0.003% 1-phenyl-2-thiourea. At 5 dpf, larvae were embedded in 2% low-melting-point agarose and imaged on a Zeiss LSM780 microscope, equipped with a 488 nm argon, 405 nm diode, and 561 nm DPSS lasers, and a ×20/1.0 NA water-dipping objective. Subsequently, larvae with sparse (1–3 ChrimsonR- and ~ 1–20 GCaMP6 expressing cells per tectum), but strong expression of actuator and indicator were transferred to the two-photon microscope for optogenetic connectivity mapping.

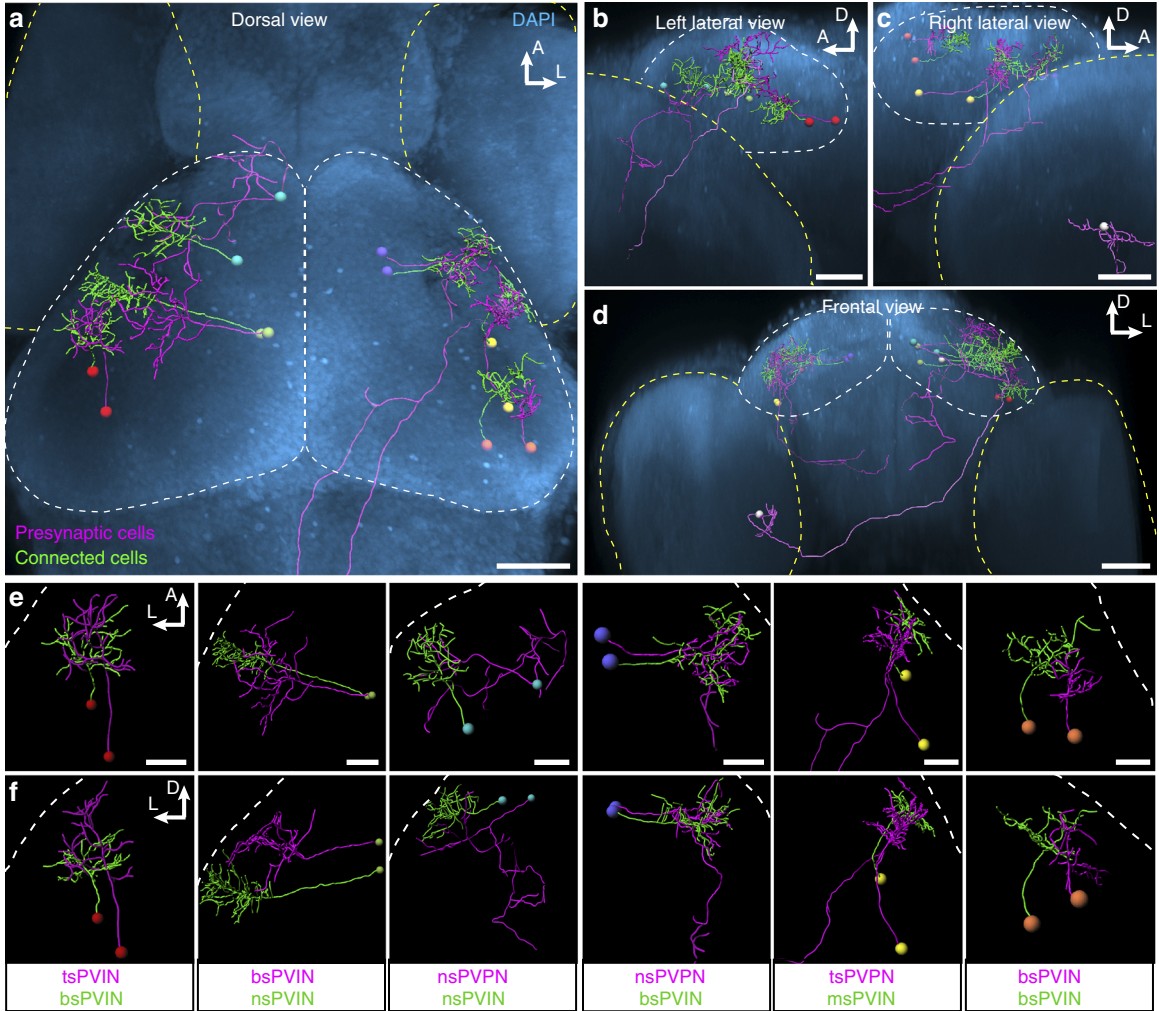

**Fig. 8** Registration of Optobow fish highlights relative anatomical positions of connected cell pairs. **a–d** Registration of six identified cell pairs (illustrated by the same cell body color) into one reference brain (outlined by DAPI). Shown are dorsal (**a**), lateral (**b** and **c**) and frontal views (**d**). Optic tecta are outlined by *white*, and eyes by *yellow dashed lines*, respectively. The RGC-PVIN cell pair (white cell bodies) described in Fig. 6 was added in **b–d**. *Scale bar*, 50 μm. **e**, **f** Dorsal (**e**) and transverse views (**f**) of single cell pairs shown in **a–d**. *White dashed lines* outline skin. Identified cell types are indicated below. *Scale bar*, 20 μm

**Two-photon microscopy.** Connectivity mapping experiments were performed on 5 dpf larvae at a commercial two-photon microscope (Femtonics, Budapest, Hungary), combining two independent Ti:Sapphire (Ti:Sa) lasers (Chameleon Ultra II, Coherent, East Hanover, NJ, USA) for photostimulation and imaging, respectively. Modulation of the intensity (MU) was performed using Pockels cells (Conoptics, Danbury, CT, USA). The photostimulation beam is, according to a custom design, deflected before the galvo-based scanhead, and its wavefront is shaped by a spatial light modulator (SLM, Hamamatsu Photonics, Hamamatsu, Japan). This additional path includes a 4× beam expansion (BE) to fill the optical window of the SLM, a half-wave plate (λ/2) to match the polarization orientation required by the crystal alignment in the SLM, and the SLM device itself working in a basic phase modulation scheme. A 400 mm Fourier lens (FL) conjugates the SLM plane, in which the phase correction is superimposed, to the first Fourier plane where the amplitude modulation is rendered. At this plane, a mirror is combined with a zero order block to suppress the residual light component that is poorly controlled by the SLM. The obtained two-dimensional amplitude distribution is reproduced at the sample plane by means of two telescopes in cascade resulting in a total magnification factor of about 1/220. The first, comprising 200 and 100 mm lenses, conjugates the Fourier plane to the back focal plane of the tube lens (TL) and includes a half wave-plate to control the direction of polarization of the photostimulation beam. The TL and the objective lens in use constitute the second telescope. Imaging and phase modulation paths are combined together in the focal plane, downstream of the Galvo Mirrors (GM), between the scan lens (SL) and the TL by means of a dichroic mirror (CO). The fluorescence collection path includes a DM670HP dichroic mirror (DM1), an infrared filter (IR), and a 563HP mirror (DM2) splitting the fluorescence light towards two GaAsP detector (PMT1/PMT2) arms (Hamamatsu Photonics, Hamamatsu, Japan) equipped with EM525/60 and

EM590/60 emission filters, respectively. An Olympus XLUMPLFL 20x/0.95 NA water-dipping objective was used.

To target single ChrimsonR-expressing cells for photostimulation, an acquired image or Z-stack and its associated metadata were imported through a Graphical User Interface (GUI). A polygonal selection tool was used to select the desired illumination patterns and to enter the corresponding Z levels of the projection. Typically, an oval-shaped circular illumination profile of 6 μm in diameter was used to activate the ChrimsonR-expressing cell body. Afterwards, the spatio-temporal protocols were designed defining the stimulation time, duration and power density. At the start of the imaging sessions, a callback is instanced triggering the execution of the photostimulation protocol. The power for photostimulation was usually around 0.2 mW μm⁻² for 760 nm or 0.5 mW μm⁻² for 1020 nm. Stimulation of ChrimsonR at 760 nm most likely results in one-photon excitation. All responses obtained using Optobow-nPA and Optobow-nPA_Syp have been confirmed both at 760 and 1020 nm photostimulations to exclude visual artifacts.

GCaMP6 signals were recorded by scanning at 920 nm in random access mode (line scans) at 250–350 Hz with a pixel size of 0.3–0.5 μm per pixel. Analysis of GCaMP6 signals was performed with a custom-made routine written in Python. Measured somatic fluorescence signals, initially processed with registration and noise subtraction algorithms, were afterwards analyzed to extract Z-scores of the samples and ΔF/F profiles considering a baseline level measured at the beginning of the recordings. Z-scores were preferred instead of ΔF/F when the intrinsic baseline noise or level made the direct ΔF/F quantification prone to generation of artifacts. For the analysis of the latency of the onset in the responses, the first sample in the raw data above a threshold defined at 4 sigma with respect to the mean in a baseline window was used to assign the time point for the response

onset. The difference between the onset time and the beginning of the stimulation phase gave the response latency. Two-sided *t*-tests were applied to obtain *P* values. Unless stated otherwise, *n* equals number of trials.

**Additional imaging and image processing.** Photoactivation of PA-GFP was either performed at the two-photon microscope (740 nm) or at the Zeiss LSM780 microscope using a 405 nm diode laser pointed at the nucleus of the cell of interest, labeled by nls-GCaMP6s. For Optobow-nPA and Optobow-nPA_Syp experiments, larvae were imaged at the confocal microscope before and after PA-GFP photo-activation. To distinguish PA-GFP from Syp-mCitrine, lambda scans were acquired using 488 nm excitation and a spectral window of 491–604 nm at 8 nm resolution. Scans were spectrally unmixed using ZEN software (black edition, v8.0; Zeiss). Image processing and filament tracing were done with Imaris software (v8.0; Bitplane).

**Immunohistochemistry and image registration.** For whole-mount immunostaining, 5 dpf zebrafish larvae were fixed in 4% paraformaldehyde at 4 °C over night. After three washes for 15 min in PBS with 0.25% Triton (PBT), larvae were incubated in 150 mM Tris-HCl, pH 9, for 5 min at room temperature followed by 15 min at 70 °C for antigen retrieval. Larvae were washed again in PBT prior to a 40 min digest with Trypsin EDTA on ice. PBT-washed larvae were blocked in 5% goat serum, 1% BSA, and 1% DMSO in PBT. Primary antibodies were added in blocking solution for 72 h. The following antibodies were used (all 1:250): anti-GFP (A10262, Life Technologies), anti-RFP (PM005, MPL), and anti-HuC/D neuronal Protein (16A11, Molecular probes). Secondary antibodies were Alexa-conjugates (Invitrogen) and added 1:300 in PBT for 48 h at 4 °C together with DAPI. Washed larvae were postfixed for 30 min in 4% paraformaldehyde at room temperature, rinsed, and thereafter transferred to 80% glycerol. Stained samples were imaged at the Zeiss LSM780 using a 25x/0.8 NA glycerol objective.

Image registration was performed using a CMTK-based GUI[43]. A 5 dpf larva stained for HuC/D was used for the template brain. The algorithm was applied to four-channel confocal stacks of Optobow-nPA expressing fish (stained as described above). Filament reconstructions of identified cell pairs were exported from live images and were manually transformed into the template brain, using the registered GFP and RFP channels of individual fish as a reference. For visualization of RGC innervation strata in the tectum, two isl2b:Gal4 UAS:GCaMP6s expressing fish were stained for GFP and HuC/D, and were co-registered into the reference brain using the HuC/D channel as a template.

**Data availability.** The data that support the findings of this study are available from the corresponding author upon reasonable request.

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

## Acknowledgements

We thank Ed Boyden for providing ChrimsonR constructs before publication. We thank Irene Arnold-Ammer, Anna Kramer and Enrico Kühn for help with cloning, Thomas Helmbrecht for help with data analysis, and Michael Kunst for help with image registration. Winfried Denk and Ruben Portugues provided comments on the manuscript. This work was funded by the Max Planck Society, the DFG (SFB 870), and the Excellence Centre for Integrated Protein Science, Munich (CIPSM). D.F. was supported by an EMBO fellowship (ALTF 104-2013).

## Author contributions

D.F., M.D.M. and H.B. designed the experiments. D.F. and M.D.M. conducted the Optobow experiments. E.L. performed the immunohistochemistry. D.F. performed the image registration. D.F. and M.D.M. analyzed the data. D.F., M.D.M. and H.B. wrote the manuscript.

## Additional information

**Competing interests:** The authors declare no competing financial interests.

