## [Peer Review File · Nature Communications]

Reviewers' Comments:

Reviewer #1 (Remarks to the Author):

This is an important paper as it provides a toolkit and some nice validation for quickly obtaining information about the direct or indirect influence of the activation of one neuron on the firing of others in the zebrafish brain. This will be valuable for exploring functional relationships of neurons across the zebrafish brain. I could be brutal and ask for some electrophysiological validation of the connectivity, but there are few who can actually do that at the moment, and this approach, as it stands, has substantial value.

I am, however, very sensitive about the tendency of people without physiological experience to misinterpret/misuse calcium imaging data to infer connectivity. The paper does say in the discussion that the evidence from this approach does not mean there is a monosynaptic connection and I applaud that. In fact, the time courses are such that these very fast fish could perform entire behaviors in the latency period, or in the 200 millisecond excitation period, so many synapses could be between the activated and the responding cell. Single cells in zebrafish can even trigger entire behaviors (that are initiated and sometimes complete in 200 ms) involving very many neurons in some cases. The authors should be even more emphatic about the fact that they cannot really tell with confidence whether the connection is direct or not from the approach they use.

Many people working on zebrafish have little circuit busting experience, so they may use the tool inappropriately. The authors do it themselves by assuming direct connections when they claim a novel discovery about en passant connectivity that depends on it being monosynaptic, when it may well not be, given the inherent temporal sloppiness of the calcium imaging. Fix that please. The abstract is not explicit about the point that MANY of the “connections” might not be monosynaptic when the delays are so long. While they may not intend it, it implies direct connection.

So, I would not support the paper's publication without some changes to the wording. They should say in the abstract that activating one cell led to calcium responses in others of varying and long latency, some of which may be monosynaptic based on cellular proximity and synaptic marker staining. And, say something to the effect that the approach can reveal potential interactions, but must be used cautiously as the influence of one cell on others may not be direct and it is difficult to be sure that it is by calcium imaging without EM or electrophysiology.

The caveats are buried in the discussion and mostly the paper uses term connection. Many people will interpret “connection” as direct. Please modify the use of the word connection everywhere in the paper to say mono or polysynaptic connections (or direct or indirect) and not use the general term connection for any interaction, however indirect.

Putting the caveats up front in the abstract will help reduce work by people who do not understand that the calcium response here are NOT necessarily, and probably not often, monosynaptic. If the authors dispute such a change, then they should show systematically that the connections (even the shortest ones they see here) are monosynaptic by physiology or EM, which are much more more definitive

ways to reveal direct connections. I predict that very many will not be, as this tool is too crude. They may not even be excitatory in the excitatory transmitter sense, because a strong inhibition can lead to rebound firing of a cell. I think the authors appreciate the issue of mono and polysynaptic as they say it in the paper, but it needs to be up front and unambiguously stated in the abstract, and anything that could lead to a misinterpretation of this expunged from the paper. The approach is STILL very useful and very important when used cautiously. I really like the toolkit, but you must do all you can to head off its misuse. Individuals will misuse it anyway, but at least you will have inundated them its limitations, so those who want to be careful, but lack the experience with actual circuit busting, know them. Also, as mentioned earlier, do not do make the mistake yourself, as you did in the paper where you write:

Further, we discovered that tectal projection neurons, which send long efferents to premotor areas or to the contralateral tectum, form en passant excitatory connections with neighbouring interneurons, or with other projection neurons (Figure 2, Figure 3, Figure 7 and Figure 8).

Your evidence here for this is not conclusive, but it is very suggestive.

I was unclear about whether the Chrimson excitation is multiphoton or not. If it is not then the problems expand because neurons along the path of the light in are potentially excited. This should be made clear, and if not two photon, then the attendant problem addressed.

Reviewer #2 (Remarks to the Author):

The approach described in this paper is a valuable technology to map the connectivity of cell types. The strategy was to target the channelrhodopsin variant ChrimsonR and the calcium sensor GCaMP6 to non-overlapping subsets of neurons in the zebrafish brain. Single cell optogenetic stimulation, using a spatial light modulator, was combined with simultaneous recording of connected cells expressing the calcium sensor. Clearly, this technique will be of interest to the community, among other approaches, such as dual microelectrode recordings of connected cells or electron microscope based reconstructions of synaptic connection (connectomics). However, the authors did not mention the technique of trans-synaptic tracers (retrograde and anterograde) which is a powerful approach that allows for mapping the connectivity of neural networks. Trans-synaptic tracing approaches should be mentioned and acknowledged in the introduction, including references.

A major remark concerns the axial confinement of the 2-photon digital holography: Is there any measurements on axial resolution? It seems that the 6um diameter mentioned at line 462 corresponds to XY dimensions. Did the authors characterize their activation point spread function? If the labeling of cells with the optogenetic tool is sparse enough to prevent activation of one cell that is located above or below the activated one, then there should be no problem. But in the abstract, the authors claim that "Optobow should be useful for the acquisition of wiring diagrams, even in dense neural circuits". It would be helpful if the authors could give some quantification concerning the axial resolution of their optical stimulation technique.

Reviewer #3 (Remarks to the Author):

Baier et al reported a novel genetic toolkit, named Optobow, for mapping functional connected neurons in zebrafish. Taking advantage of Gal4-UAS together with Cre/lox, in a similar fashion with “brainbow”, they target the optogenetic actuator ChrimsonR and calcium sensors into separate subsets of neurons. Using photostimulation and imaging, they are able to identify pairwise connected neurons in larval zebrafish tectum. They further demonstrate the utility of three types of Optobows in mapping connected neurons in densely packed areas with morphological details in tectum, as well as for paired neurons over long distances. A broad application of this toolkit will have no doubt to add on existing tools for circuitry mapping and most likely new information will be revealed. However, concerns, mostly focusing on experimental design and quantification as listed in below, need to be addressed before fully evaluating the usefulness of this toolkit.

Major concerns:

1. The author’s approach to sparsely label subsets of neurons with three genes included in the design of Optobow is confounding (Schematic representation in Fig 1A, 2A, 3A, 4A). Based on current design, specifically regarding the orientation of loxP and markers including mCitrine, ChrimR and GCaMP, as indicated in schematic drawing in Fig 1A, 2A, 3A, 4A, it is not possible to achieve distinct labeling of three subset populations with each of genes in Optobow. If we ignore the schematic drawing, the method part didn’t provide any relevant information about how this sparse, separate labeling was achieved. This has to be straight out before judging the feasibility of this approach. Also this information is absolutely needed to understand the yield and scalability of this approach in mapping connected neurons.

2. The rigor of experimental approach is relatively weaker.

2.1 In Fig1-4, the quantification of expression pattern of Optobow is lacking. For example, what is efficacy of such labeling strategy? Percentage of cells labeled? How many neurons can be labeled in each subset of populations?

2.2 The characterization of Optobow, for example, expression level, chronic expression pattern, how the expression alter the original circuitry or any altered phenotype and morphological changes were observed, were not sufficient or at least should be discussed.

2.3 The validation of this all-optical mapping method for identifying paired connected neurons is preferred, for example subcellular evidence to show synapses identification. For example, the evidence by spectral unmixing is not convincing to identify punctae that are less than 1 μ M in distance.

2.4 In terms of 2P calcium imaging, what is the spatial resolution and field of view with 350Hz scanning? whether it is necessary to use such a high scanning speed? This again raised questions regarding the yield and scalability of this optical approach. The difference of latency of calcium response is also likely due to the altered calcium dynamics in individual cells as a result of heterogeneous level of expression of calcium sensors.

3. It is not clear whether such toolkit can be used in other model systems, for example, drosophila or even rodents.

Response to reviewers

Reviewer #1 (Remarks to the Author):

This is an important paper as it provides a toolkit and some nice validation for quickly obtaining information about the direct or indirect influence of the activation of one neuron on the firing of others in the zebrafish brain. This will be valuable for exploring functional relationships of neurons across the zebrafish brain. I could be brutal and ask for some electrophysiological validation of the connectivity, but there are few who can actually do that at the moment, and this approach, as it stands, has substantial value.

We thank the reviewer for this positive evaluation.

I am, however, very sensitive about the tendency of people without physiological experience to misinterpret/misuse calcium imaging data to infer connectivity. The paper does say in the discussion that the evidence from this approach does not mean there is a monosynaptic connection and I applaud that. In fact, the time courses are such that these very fast fish could perform entire behaviors in the latency period, or in the 200 millisecond excitation period, so many synapses could be between the activated and the responding cell. Single cells in zebrafish can even trigger entire behaviors (that are initiated and sometimes complete in 200 ms) involving very many neurons in some cases. The authors should be even more emphatic about the fact that they cannot really tell with confidence whether the connection is direct or not from the approach they use. Many people working on zebrafish have little circuit busting experience, so they may use the tool inappropriately.

We appreciate the reviewer's concern about a potential misunderstanding of direct or indirect connectivity. We carefully rephrased several sections in the text and reconsidered the use of the term "connection" and "connectivity" (also see below). We made the following changes in the text:

- Results, line 95: deleted "postsynaptic"
- Results, line 185: replaced "postsynaptic" by "reporter"
- Results, line 208: added "of one functional circuit"
- Results, line 239: rephrased sentence
- Discussion, line 349: replaced "postsynaptic" by "downstream"
- Legend Figure 3: Title changed from "postsynaptically active cells" to "functionally connected cells"
- Legend Figure 3E and F: replaced "postsynaptic cell" by "cell#1"
- Legend Figure 4F: replaced "postsynaptic" by "connected"
- Figure 4F: replaced "postsynaptic cell" by "connected cell"
- Figure 4F: replaced "potential synapses" by "synaptic terminals"
- Figure 7A: replaced "postsynaptic cell" by "connected cell"
- Figure 8A: replaced "postsynaptic cells" by "connected cells"
- Legend Supplementary Movie 2: replaced "first postsynaptic" and "second postsynaptic cell" by "connected cell#1" and "connected cell#2"

The authors do it themselves by assuming direct connections when they claim a novel discovery about en passant connectivity that depends on it being monosynaptic, when it may well not be, given the inherent temporal sloppiness of the calcium imaging. Fix that please.

We toned down our statement in the Results (line 281) by replacing "form en passant excitatory connections with" by "are directly or indirectly connected to".

The abstract is not explicit about the point that MANY of the "connections" might not be monosynaptic when the delays are so long. While they may not intend it, it implies direct connection. So, I would not support the papers publication without some changes to the wording. They should say in the abstract that activating one cell led to calcium responses in others of varying and long latency, some of which may be monosynaptic based on cellular proximity and synaptic marker staining.

We rewrote the relevant paragraph in the abstract.

And, say something to the effect that the approach can reveal potential interactions, but must be used cautiously as the influence of one cell on others may not be direct and it is difficult to be sure that it is by calcium imaging without EM or electrophysiology.

The caveats are buried in the discussion and mostly the paper uses term connection. Many people will interpret "connection" as direct. Please modify the use of the word connection everywhere in the paper to say mono or polysynaptic connections (or direct or indirect) and not use the general term connection for any interaction, however indirect.

We reviewed the use of the term connection:

- line 177: added "direct or indirect"
- line 197: added "directly or indirectly"
- line 259: replaced "interconnected" by "directly or indirectly connected"
- line 294: replaced "individual interconnected" by "belonging to individual functional circuits"
- line 312: added "direct or indirect"

Putting the caveats up front in the abstract will help reduce work by people who do not understand that the calcium response here are NOT necessarily, and probably not often, monosynaptic. If the authors dispute such a change, then they should show systematically that the connections (even the shortest ones they see here) are monosynaptic by physiology or EM, which are much more definitive ways to reveal direct connections. I predict that very many will not be, as this tool is too crude. They may not even be excitatory in the excitatory transmitter sense, because a strong inhibition can lead to rebound firing of a cell.

We added a sentence in the discussion (line 349) to make the reader aware of these rare but possible events.

I think the authors appreciate the issue of mono and polysynaptic as they say it in the paper, but it needs to be up front and unambiguously stated in the abstract, and anything that could lead to a misinterpretation of this expunged from the paper. The approach is STILL very useful and very important when used cautiously. I really like the toolkit, but you must do all you can to head off its misuse. Individuals will misuse it anyway, but at least you will have inundated them its limitations, so those who want to be careful, but lack the experience with actual circuit busting, know them. Also, as mentioned earlier, do not do make the mistake yourself, as you did in the paper where you write:

Further, we discovered that tectal projection neurons, which send long efferents to premotor areas or to the contralateral tectum, form en passant excitatory connections with neighbouring interneurons, or with other projection neurons (Figure 2, Figure 3, Figure 7 and Figure 8).

We rephrased this paragraph (see above).

Your evidence here for this is not conclusive, but it is very suggestive.

I was unclear about whether the Chrimson excitation is multiphoton or not. If it is not then the problems expand because neurons along the path of the light in are potentially excited. This should be made clear, and if not two photon, then the attendant problem addressed.

While Chrimson excitation at 1020 nm is due to a two-photon effect, stimulation at 760 nm most likely results in one-photon excitation (see Supplementary Figure 4B). All responses obtained using Optobow-nPA and Optobow-nPA_Syp have been confirmed both at 760 nm and 1020 nm photostimulations. We added this information to the methods section (line 436). Additionally, a detailed characterization of the spatial specificity of our holographic photostimulation has been added (see response to reviewer 2). Further, for all experiments, only fish with sparse expression of Chrimson have been selected (1-3 Chrimson-expressing cells per tectum, see Methods, line 398), to exclude unintentional activation of neighboring cells.

Reviewer #2 (Remarks to the Author):

The approach described in this paper is a valuable technology to map the connectivity of cell types. The strategy was to target the channelrhodopsin variant ChrimsonR and the calcium sensor GCaMP6 to non-overlapping subsets of neurons in the zebrafish brain. Single cell optogenetic stimulation, using a spatial light modulator, was combined with simultaneous recording of connected cells expressing the calcium sensor.

Clearly, this technique will be of interest to the community, among other approaches, such as dual microelectrode recordings of connected cells or electron microscope based reconstructions of synaptic connection (connectomics).

We thank the reviewer for this positive evaluation.

However, the authors did not mention the technique of trans-synaptic tracers (retrograde and anterograde) which is a powerful approach that allows for mapping the connectivity of neural networks. Trans-synaptic tracing approaches should be mentioned and acknowledged in the introduction, including references.

The reviewer's point is well taken. We now added transsynaptic tracing technologies and their references to the introduction (line 63).

A major remark concerns the axial confinement of the 2-photon digital holography: Is there any measurements on axial resolution? It seems that the 6 μ m diameter mentioned at line 462 corresponds to XY dimensions. Did the authors characterize their activation point spread function? If the labeling of cells with the optogenetic tool is sparse enough to prevent activation of one cell that is located above or below the activated one, then there should be no problem. But in the abstract, the authors claim that "Optobow should be useful for the acquisition of wiring diagrams, even in dense neural circuits". It would be helpful if the authors could give some quantification concerning the axial resolution of their optical stimulation technique.

The reviewer is interested in the important issue of spatial specificity of our holographic photostimulation. We addressed this issue by adding a new Supplementary Figure 4 to this manuscript. It comprises a characterization of the axial and lateral resolution of our holographic stimulation spot (panel A). In addition, we tested the lateral selectivity in a situation, where several neighboring and overlapping neurons are co-expressing ChrimsonR-tagRFP and nls-GCaMP6s (panel B). Even though the effectiveness of photostimulation decreases with distance, these results emphasize the need for sparse labeling, which can be achieved using our Optobow approach. Light-microscopic acquisition of wiring diagrams in dense neural circuits can only be achieved by a sparse labeling approach.

While our paper is mostly concerned with the genetic technology, the issue of cellular resolution is addressed in depth in another manuscript that is concerned with the optical techniques (Dal Maschio et al., in revision). Several independent experiments were performed to compare the actual with the desired illumination profiles *in vivo* and to ensure proper calibration of the system. We have included a figure from that manuscript for the reviewer's perusal:

(C) Axial and lateral average projections of a photoactivation volume obtained *in vivo* with a target pattern 6 μ m in diameter. The paGFP profiles (FWHM: $5.84 \pm 0.56 \mu$ m laterally, and $7.86 \pm 1.2 \mu$ m axially), photoactivated at 750 nm with a power of $0.25 \text{ mW}/\mu\text{m}^2$ at the sample, closely match the typical cell diameter. **(E)** Parallel neuronal stimulation *in vivo*. A small network of neurons is imaged during repeated simultaneous photostimulation of eight targeted neurons (in blue). Photostimulations are indicated by orange vertical lines, and the traces for photostimulated neurons are shown in blue. **(F)** Effective resolution of photostimulation. To measure lateral resolution, targeted cells were photostimulated while nearby neurons in the same imaging plane were recorded. Each grey dot represents a non-targeted cell, with $\Delta F/F$ normalized relative to the response in

the targeted cell. An exponential decay fit is shown (black line) with 95% confidence interval (bootstrap, shown in grey). The thin dashed line shows the minimum distance between cells in this region of the brain. To measure axial (Z) resolution, the imaging plane was shifted axially from the target, either with the ETL or by adjusting the holographic pattern. Induced $\Delta F/F$ is normalized relative to the photostimulated target cell. In blue, an exponential fit is shown, and error bars indicate the S.E.M. Scale bars are 10 μm .

Panel (C) shows the axial and lateral confinement of photostimulation using a circular illumination profile of 6 μm in diameter and a power density equal or similar to the ones used for ChrimsonR in the study at hand. We observed that PA-GFP, which is expressed in all cells of the densely packed optic tectum, was precisely photoactivated in a single cell. We conclude that light scattering by the tissue did not substantially distort the photostimulation profile. This experiment can only serve as an approximation for our optogenetic targeting specificity, since PA-GFP and ChrimsonR may have different 2P cross-sections.

Panel (E), using in this case ChR2 and GCaMP6 co-expressed in the same cells, shows the result of an experiment in which eight illumination spots (each 6 μm in diameter) were generated to stimulate eight different neurons. Non-targeted ChR2-expressing cells in the immediate vicinity did not show any calcium response. Activity in non-targeted cells was sporadically detected and is likely resulting from extension of neuronal processes into the photostimulated regions (green asterisks). A "cross-activation length constant" was calculated in panel (F), in which the lateral and axial decay profiles of normalized $\Delta F/F$ response amplitude are plotted versus the distance from the targeted neurons.

In conclusion, we are fairly confident that our 2P holographic method enables spatially defined excitation of sparsely labeled Chrimson cells in our experiments.

Reviewer #3 (Remarks to the Author):

Baier et al reported a novel genetic toolkit, named Optobow, for mapping functional connected neurons in zebrafish. Taking advantage of Gal4-UAS together with Cre/lox, in a similar fashion with "brainbow", they target the optogenetic actuator ChrimsonR and calcium sensors into separate subsets of neurons. Using photostimulation and imaging, they are able to identify pairwise connected neurons in larval zebrafish tectum. They further demonstrate the utility of three types of Optobows in mapping connected neurons in densely packed areas with morphological details in tectum, as well as for paired neurons over long distances. A broad application of this toolkit will have no doubt to add on existing tools for circuitry mapping and most likely new information will be revealed. However, concerns, mostly focusing on experimental design and quantification as listed in below, need to be addressed before fully evaluating the usefulness of this toolkit.

We appreciate the time the reviewer took for evaluating our work. We believe we have addressed each of the concerns below.

Major concerns:

1. The author's approach to sparsely label subsets of neurons with three genes included in the design of Optobow is confounding (Schematic representation in Fig 1A, 2A, 3A, 4A). Based on current design, specifically regarding the orientation of loxP and markers including mCitrine, ChrimR and GCaMP, as indicated in schematic drawing in Fig 1A, 2A, 3A, 4A, it is not possible to achieve distinct labeling of three subset populations with each of genes in Optobow. If we ignore the schematic drawing, the method part didn't provide any relevant information about how this sparse, separate labeling was achieved. This has to be straight out before judging the feasibility of this approach. Also this information is absolutely needed to understand the yield and scalability of this approach in mapping connected neurons.

The reviewer likely misunderstood the design and the goal of the Optobow strategy. As stated in the results (line 115), the approach leads to stochastic and unbiased labeling of random cells within one neuronal population (defined by the Gal4 driver). Thus, provided that the driver line leads to expression in different cell types, by chance, one can achieve their distinct and sparse labeling. The yield of this approach for finding connected neurons will depend on the interconnectivity within the population of interest and can be adjusted by increasing/decreasing actuator and indicator expression by varying Cre concentration and Optobow constructs (see Discussion, line 354).

2. The rigor of experimental approach is relatively weaker.

2.1 In Fig1-4, the quantification of expression pattern of Optobow is lacking. For example, what is efficacy of such labeling strategy? Percentage of cells labeled? How many neurons can be labeled in each subset of populations?

As mentioned above, the percentage of cells labeled in the population of interest will strongly depend on the variegation of *UAS:Optobow* constructs and the concentration of Cre (also see Results, line 113). In our study, we screened for fish larvae showing sparse expression, i.e. 1-3 Chrimson- and ~1-20 GCaMP-expressing cells per tectum. We added this information to the Methods section (line 398).

2.2 The characterization of Optobow, for example, expression level, chronic expression pattern, how the expression alter the original circuitry or any altered phenotype and morphological changes were observed, were not sufficient or at least should be discussed.

The expression levels and the temporal expression pattern will depend on the Gal4 driver line. In our study, we used the *Gal4s1013t* enhancer trap line, previously characterized to drive broad and strong expression in the tectum (Scott and Baier, *Front Neural Circuits*, 2009). For all Optobow experiments, we selected for fish showing strong expression of actuator and indicator (methods, line 399).

We have not observed toxicity or any phenotype evoked by the effector proteins used. We have generated several transgenic fish lines, which stably express *UAS:ChrimsonR-tagRFP* (this study) or *UAS:nlsGCaMP6+PA-GFP*. All of these lines are kept either together with *Gal4s1101t* or with *elav13:Gal4*, both leading to strong, life-long pan-neuronal expression. Adult carriers of these transgenes are viable and fertile, and their behavior appears unaffected, indicating that cell health is not compromised. This information can be found in the Methods section (line 382).

2.3 The validation of this all-optical mapping method for identifying paired connected neurons is preferred, for example subcellular evidence to show synapses identification. For example, the evidence by spectral unmixing is not convincing to identify punctae that are less than 1 μ M in distance.

We weakened our statement for identifying directly connected neurons using synaptic marker stainings (results, line 239).

Subcellular evidence, i.e. electron microscopy of synaptic vesicles between two previously identified cells in this dense circuitry is clearly beyond the scope of the current study. We are not aware of correlative EM analyses of the synapses of connected cell pairs. These experiments are unprecedented and would take extensive time and resources, which would fill an additional paper.

2.4 In terms of 2P calcium imaging, what is the spatial resolution and field of view with 350Hz scanning? whether it is necessary to use such a high scanning speed? This again raised questions regarding the yield and scalability of this optical approach. The difference of latency of calcium response is also likely due to the altered calcium dynamics in individual cells as a result of heterogeneous level of expression of calcium sensors.

The field of view with scanning at 350 Hz is a line with a pixel size of 0.3-0.5 μ m/pixel (10-15 pixel/cell). This information has been added to the Methods section (line 440). This high scanning speed is needed, if subsequent characterization of GCaMP response latencies is desired. Additionally, line scanning, in contrast to raster scanning, prevents unintentional activation of PA-GFP in the same cell or of the optogenetic actuator in nearby cells. Further, it drastically increases the yield of the approach as many more GCaMP cells can be monitored simultaneously at sufficient temporal resolution.

Expression levels of the calcium indicator can indeed vary from cell to cell, due to the properties of the Gal4/UAS system. However, latency differences due to varying expression levels cannot explain our finding that cells with longer response latencies statistically do not show overlapping neurites with the stimulated cell.

3. It is not clear whether such toolkit can be used in other model systems, for example, drosophila or even rodents.

The general "Brainbow" expression system (combination of Cre and mutually exclusive pairs of lox sites), as well as the "stand-alone" genetic components Chrimson, PA-GFP, and nuclear-localized GCaMP have already been successfully applied in *Drosophila* (Hempel et al., *Nat Meth*, 2011; Klapoetke et al., *Nat Meth*, 2014; Ruta et al., *Nature*, 2010; Weislogel et al., *Sci Signal*, 2013) and mouse (Livet et al., *Nature*, 2007; Klapoetke et al., *Nat Meth*, 2014; Lien and Scanziani, *Front Neural Circuits*, 2011; Bengtson et al., *Biophys J*, 2010). The concept of using a combination of red-shifted actuators and GCaMP has been applied to acute brain slices and

in vivo in mouse (Packer et al., *Nat Meth*, 2014; Rickgauer et al., *Nat Neurosci*, 2014), and most recently in *Drosophila* (Sen et al., *Curr Biol*, 2017). While the combination of nuclear-localized GCaMP and PA-GFP has not been published so far, our collaborators from the lab of Tobias Bonhoeffer recently achieved functional co-expression of both components in mouse brain slices (T. Rose, pers. comm.). We think that these indications make it fair to discuss a potential adaptation of our Optobow toolbox to other genetic model organisms, like fruit fly or mouse (Discussion, line 331). Particularly for applications in mouse, one could imagine viral delivery of Optobow under control of a pan-neuronal promoter, while making use of the huge collection of available Cre lines to direct recombination to a cell population of interest.

Reviewers' Comments:

Reviewer #1 (Remarks to the Author):

I liked the original version of the paper and mainly was concerned about the implication of monosynaptic connections when many might be polysynaptic. The authors have done a good job of making clear that they cannot conclusively tell whether they are monosynaptic or polysynaptic. I think this is indeed a powerful approach even without that ability to resolve how many synapses are in between. I am happy with the revisions.

Reviewer #2 (Remarks to the Author):

Thank you for the revisions. It was a pleasure to read the revised text. I also appreciate that the authors have included a supplementary figure showing the axial and lateral resolution of their holographic stimulation.

Reviewer #3 (Remarks to the Author):

The authors have addressed most of my concerns. Even though they tuned down the conclusion about “directly connected neurons”, I still think more control experiments such as post-hoc structural analysis of functional paired neurons at subcellular level (subcellular distribution of synapses) is absolutely necessary to validate the results obtained using Optobow. This is necessary as electrophysiology is not an option here in fish. However, I do agree with the authors that it can be a follow-up study. In addition, I still think applications of Optobow in rodent may be limited, unless more evidence can be provided. Though I am aware of limitations and usefulness of Optobow can offer, there is no doubt that an immediate dissemination of Optobow fish lines will potentially benefit labs studying neural circuitry mechanisms using zebrafish models. Therefore, I recommend acceptance and publication of this work.